# Identification of the Factor That Leads Human Mesenchymal Stem Cell Lines into Decellularized Bone

**DOI:** 10.3390/bioengineering9100490

**Published:** 2022-09-21

**Authors:** Anri Koyanagi, Iichiroh Onishi, Karin Muraoka, Ikue Sato, Shingo Sato, Tsuyoshi Kimura, Akio Kishida, Kouhei Yamamoto, Masanobu Kitagawa, Morito Kurata

**Affiliations:** 1Department of Comprehensive Pathology, Graduate School of Medical and Dental Sciences, Tokyo Medical and Dental University, Tokyo 113-8510, Japan; 2Faculty of Health Science Technology, Bunkyo Gakuin University, Tokyo 113-8668, Japan; 3Center for Innovative Cancer Treatment, Tokyo Medical and Dental University, Tokyo 113-8510, Japan; 4Department of Material-Based Medical Engineering, Tokyo Medical and Dental University, Tokyo 101-0062, Japan

**Keywords:** bone marrow niche, mesenchymal stem cells, decellularized bone marrow, CRISPR SAM activation library

## Abstract

Hematopoiesis is maintained by the interaction of hematopoietic stem cells (HSCs) and bone marrow mesenchymal stem cells (MSCs) in bone marrow microenvironments, called niches. Certain genetic mutations in MSCs, not HSCs, provoke some hematopoietic neoplasms, such as myelodysplastic syndrome. An in vivo bone marrow niche model using human MSC cell lines with specific genetic mutations and bone scaffolds is necessary to elucidate these interactions and the disease onset. We focused on decellularized bone (DCB) as a useful bone scaffold and attempted to induce human MSCs (UE7T-9 cells) into the DCB. Using the CRISPR activation library, we identified *SHC4* upregulation as a candidate factor, with the *SHC4* overexpression in UE7T-9 cells activating their migratory ability and upregulating genes to promote hematopoietic cell migration. This is the first study to apply the CRISPR library to engraft cells into decellularized biomaterials. *SHC4* overexpression is essential for engrafting UE7T-9 cells into DCB, and it might be the first step toward creating an in vivo human–mouse hybrid bone marrow niche model.

## 1. Introduction

The bone marrow niche is defined as a local tissue microenvironment that maintains and regulates hematopoietic stem cells (HSCs). Many previous studies have suggested that mesenchymal stem cells (MSCs) and vascular endothelial cells maintain HSCs and regulate their proliferation and differentiation in the bone marrow niche [1,2,3,4]. The bone marrow niche has been rapidly elucidated in recent years by next-generation sequencing and cell analysis techniques [5]. 

In order to analyze the bone marrow niche, many studies have attempted to develop humanized hematopoietic models, such as patient-derived xenograft (PDX) and humanized ossicle (hOss), which are small human bone organs with structures and functions similar to mouse bones [6,7]. PDX models need to collect patient-derived cells and have difficulty preparing cells with genetically equivalent conditions. HOss models require bone marrow-derived MSCs (BM-MSCs) to differentiate osteogenic lineages and be seeded onto scaffold material prior to the implantation of HSCs.

In recent years, various biomaterials have been generated by the recellularization of cells into the skeleton of decellularized extracellular matrices (dECM) [8]. We focused on porcine decellularized bone (DCB) as a scaffold, which is categorized as a dECM. Hashimoto et al. reported that the subcutaneous transplantation of DCBs into rats established hematopoietic tissue in DCBs without provoking an immune response [9]. According to Nakamura et al., sodium-dodecyl-sulfate (SDS)-treated DCB, stripped of reticular and adipose tissues, lacks long-term HSCs in vivo, which can be found in high hydrostatic pressure-treated DCBs, which were used in our study [10]. Unlike the hOss model, there is no need to wait for seeded BM-MSCs to differentiate into osteogenic lineages. DCB with reticular and adipose tissue is the scaffold of choice to create a microenvironment for hematopoiesis.

Myelodysplastic syndrome (MDS) is a hematological neoplasm in which abnormal HSCs with genetic mutations proliferate. Recent studies have reported that more MSCs are in contact with HSCs in MDS [11]. Genetic abnormalities, such as the *DICER1* mutation in MSCs, not HSCs, may induce MDS [11,12,13]. Mian et al. cultured MDS patient-derived HSCs and MSCs in gelatin sponge, transplanted them into mice, and showed a sustained human MDS niche in vivo [14]. Patient-derived samples are heterogeneous for each patient. To analyze the disease onset of human MDS in the bone marrow niche, it is necessary to prepare hMSCs with identical genetic abnormalities, induct them to certain scaffolds, and transplant them to mice. We hypothesize that DCB is an appropriate biomaterial because it enables the analysis of the interactions of hMSCs and mouse hematopoietic cells without adding any cytokines or differentiation factors.

We hypothesized that if genetically engineered human MSCs (hMSCs) were introduced and grown on DCBs, the subcutaneous implantation of these DCBs in mice could recapitulate the human–mouse hybrid bone marrow niche that could support the interaction of hMSCs and mouse hematopoietic cells.

The induction of hMSCs into DCB was difficult. We first cultured UE7T-9 cells (human bone marrow-derived MSC cell line) on DCB, but failed to achieve intraosseous expansion (Appendix A). Nakamura et al. demonstrated that culturing human bone marrow stem cells with DCB at 4 °C reduced their cell adhesive capacity and allowed them to infiltrate the reticular tissue within DCB in the short term. However, the distance of cell infiltration decreases with the days in culture, suggesting that the long-term engraftment of human bone marrow stem cells into DCB is very difficult [15]. Introducing hMSCs into DCBs is the very important first step toward establishing a hybrid bone marrow niche in vivo.

We used the clustered, regularly interspaced short palindromic repeats Synergistic Activation Mediator library (CRISPR SAM library). The CRISPR library is a gene-editing technology that has been used for research worldwide [16,17]. The CRISPR SAM library enables the overexpression of target genes using dCas9 (Cas9 with the catalytic domain deactivated) and MS2-p65-HSF1 protein [18]. Using this system, the CRISPR SAM library can upregulate genes randomly induced by sgRNAs and identify the genes by sequencing the selected sgRNA after selection.

To recapitulate the infiltration of hMSCs into DCB, we sought to identify the factors that promote the progression of UE7T-9 cells into DCB using the CRISPR SAM Library. We validated the identified genes that would promote UE7T-9 cell integration in DCB.

## 2. Materials and Methods

### 2.1. Cell lines and Cell Culture

Human mesenchymal stem cells (UE7T-9 cells) were purchased from the Japanese Cancer Research Bank (JCRB, Osaka, Japan) cell bank in 2014 (https://cellbank.nibiohn.go.jp/~cellbank/cgi-bin/search_res_det.cgi?ID=3821, accessed on 2 May 2022). Human embryonic kidney cells (HEK 293T) were also purchased from JCRB. UE7T-9 cells and HEK 293T cells were maintained with Dulbecco’s modified Eagle’s medium (D-MEM, Fujifilm) supplemented with 10% fetal bovine serum (Thermo Fisher Scientific, Walthum, MA, USA) and penicillin/streptomycin (Thermo Fisher Scientific, Walthum, MA, USA), incubated at 37 °C with 5% CO_2_. D-MEM was used as the culture medium throughout the experiments.

### 2.2. Transfection of SAM Library into Cell Lines

To transduce the lentivirus, HEK 293T cells (5 × 10^5^ cells/well) were seeded in a six-well cell culture plate for one day before transfection. The next day, 3 µg of lentivirus plasmid was transfected with 1 µg of pMD2.G and 2 µg of pCMV using Lipofectamine™ 3000 Reagent (Thermo Fisher Scientific, Walthum, MA, USA), according to previous work [19]. Twelve hours after transfection, the medium was changed to fresh DMEM. The virus supernatant was harvested 48 h after transfection and then filtered with Millex™ -HP 0.45 µm (Merck, Kenilworth, NJ, USA). The target HEK 293T cells (5 × 10^5^ cells/well) and UE7T-9 cells (2 × 10^5^ cells/well) were seeded in a six-well cell culture plate one day prior to transduction. The cells were transduced with this lentivirus supernatant with 5 µg/mL polybrene (Sigma-Aldrich, MO, USA). Transduction was performed using spin infection followed by 30 min of centrifugation at 4680 rpm. The lentiviral plasmids were lentiMPHv2 (Addgene, #89308, MA, USA), lenti-dCAS-VP64_blast (Addgene, #61425), and lentiSAMv2 (Addgene, #61597) [17]. First, lentiMPHv2 was transduced into HEK 293T and UE7T-9 cells, and transduced cells were selected using hygromycin B for 2 weeks. Then, lenti-dCAS9-VP64 was induced in MPH-expressing HEK 293T cells and UE7T-9 cells, and selected by Blasticidin (10 μg/mL, Thermo Fisher Scientific, Walthum, MA, USA) for 2 weeks. MPH-dCas9-VP64-expressing HEK 293T and UE7T-9 cells were applied to further experiments. The CRISPR-activated library, lenti-SAMv2, was also transduced into MPH-dCas9-VP64-expressing UE7T-9 cells. Although the multiplicity of infection (MOI) is usually set to 1 or less, in this study, we allowed duplicate infections (MOI > 1) because our goal was to invade DCB. We then performed selections with zeocin (300 μg/mL, Thermo Fisher Scientific, Walthum, MA, USA) for 2 weeks.

### 2.3. DCB Formation

Bone preparation was performed as described in a previous report [8,13]. Fresh pig femurs and ribs were obtained from a local slaughterhouse (Tokyo Shibaura Shipbuilding, Tokyo, Japan). Femurs (compact bones) and costae (trabecular bones) were cut into small pieces (4 × 4 × 3 mm) and washed in phosphate buffered saline (PBS) (Invitrogen, Tokyo, Japan) containing penicillin (100 units/mL) and streptomycin (100 μg/mL). The bone/marrow was filled with PBS and sealed in a plastic pack to prevent implosion and leakage during the pressure application. The cells were dissociated by hydrostatic pressurization at 980 MPa and 30 °C for 10 min using a cold isostatic pressurizer (Dr. CHEF, Kobe Steel, Ltd., Hyogo, Japan). Pressurization and decompression were performed at 65.3 MPa/min, and propylene glycol was used as the permeate. The cells were then washed with saline containing DNase I (0.2 mg/mL) (Roche Diagnostics, Tokyo, Japan) antibiotic for 3 weeks at 37 °C with continuous slow shaking, and then treated with 80% v/v EtOH at 37 °C for 7 days. After washing and shaking with PBS again, they were stored at 4 °C.

### 2.4. Cell Culture on DCB and Histological Evaluation

To evaluate infiltration into DCB, HEK 293T cells and UE7T-9 cells (1.0 × 10⁵ cells/well) were cultured on DCB for 3 or 8 weeks. After culture, the cells were fixed in 10% neutral buffered formalin for 1 day and demineralized in 18.5% EDTA solution (Pharma, Tokyo, Japan) for 1 day. Then paraffin-embedded sections were prepared, and hematoxylin-eosin staining of thin sections (3 μm) was performed. We photographed stained sections with a NanoZoomer S210 virtual slide scanner (Hamamatsu Photonics, Shizuoka, Japan) and measured distances using NDP.view2 (Hamamatsu Photonics K.K., Hamamatsu, Japan).

### 2.5. DNA Sequence Analysis

Paraffin-embedded sections prepared as described in the section “Cell culture on DCB and histological evaluation” were thinly sliced at 10 μm. For five sections of DCB with surface cells removed, we added 0.5 mL of DEXPAT (TaKaRa, Shiga, Japan), heated at 100 °C for 10 min, and centrifuged at 4 °C at 12,000 rpm for 10 min. The aqueous layer portion was collected, avoiding the paraffin thin film, and then purified with Wizard SV Gel & PCR Clean-up System (Promega, Madison, WI, USA ). For gRNA detection, PCR by the KOD-FX (TOYOBO, Osaka, Japan) method was then performed using SAM 6025F (tttcttgggtagtttgcagt) and SAM 6190R (cctcatgttggcctagctctct) as primers. The PCR products were purified and introduced into Zero Blunt TOPO vector using Zero Blunt TOPO PCR Cloning Kit (Thermo Fisher Scientific, Walthum, MA, USA). We performed DNA sequence analysis (Sanger method), identified gRNA, and searched for the corresponding gene in the gRNA list of the CRISPR SAM Library.

(https://view.officeapps.live.com/op/view.aspx?src=https%3A%2F%2Fmedia.addgene.org%2Fcms%2Ffiler_public%2Fea%2Fc2%2Feac2ada9-2a15-45f8-9f3d-b0782df0c695%2Fsam_human_lbrary_annotated.xlsx&wdOrigin=BROWSELINK, accessed on 8 February 2022).

### 2.6. Creation of SHC4-Overexpressing Cell Lines

The gene fragments for the coding region of *SHC4* with SAL1 and HIND3 restriction enzyme sites at each end were purchased from Integrated DNA Technologies (Appendix A for sequence diagram), integrated into 150 ng pENTR221 vectors, and incubated at room temperature for 4 h. pENTR221-GFP was used as a control. Next, it was replaced with 150 ng of plenti CMV Puro DEST vector (Plasmid #17452, Addgene) by LR reaction usingGateway^®^ LR Clonase (Thermo Fisher Scientific, Walthum, MA, USA) and incubated at room temperature for 4 h. We collected the plasmid using Wizard^®^ Plus SV Minipreps DNA Purification Systems (Promega, Madison, WI, USA) and confirmed the presence of the target bands with electrophoresis. We created two types of plasmids (plenti CMV GFP puro and plenti CMV *SHC4* puro). We then performed lipofection as described in the section “Transfection of SAM library into cell lines”, and we collected viral supernatant, which was used to infect UE7T-9 cells (2.0 × 10⁵ cells/well each). We adjusted the culture volume to 25 μg/mL/well, and then performed selection using puromycin.

### 2.7. Western Blot

We used an SDS buffer (25% of 0.125M Tris-HCl (pH 6.8), 20% of glycerol, 4% of SDS, and 10% of 2-mercaptoethanol, with bromophenol blue included) to pellet cell pellets, extract proteins, and degrade DNA by sonication in ice-cold conditions. The protein extracts were separated in Mini-PROTEAN TGX Stain-Free gels (BIO RAD, CA, USA) by electrophoresis (60 mA, 40 min) and transferred to TransBlot Turbo Mini-size PVDF membranes (BIO RAD). The membranes were blocked with Bullet Blocking One (Nacalai Tesque, Kyoto, Japan). The antibodies we used were as follows: primary antibodies—*SHC4* polyclonal antibody (rabbit, 75 kDa) (Sigma-Aldrich, Mizulli, USA) and β-actin monoclonal antibody (rabbit; 45 kDa) (Cell signaling technology, #4790, MA, USA) diluted 1000×, reacted at 4 °C in overnight. The membranes were washed with TBS-T, and the horseradish peroxidase (HRP)-conjugated anti-rabbit antibody (GE Healthcare, Chicago, IL, USA) diluted 5000× was reacted with membrane for 1 h. The membranes were washed with TBS-T three times before being exposed to the ClarityTM Western ECL Substrate (Bio Rad Laboratories, Hercules, CA, USA) as a chromogenic substrate for HRP. The protein bands were detected using a ChemiDoc MP ImageLab (Bio Rad Laboratories, Hercules, CA, USA).

### 2.8. MTS Assay

An MTS assay was performed to evaluate cell proliferation. We cultured UE7T-9 cells at 0.5 × 10⁴ cells/well in 96-well plates. At 24, 48, and 72 h of culture, we added 20 µL of MTS (2 mg/mL) (Promega, Madison, WI, USA )/PMS (0.92 mg/mL) (Promega, Madison, WI, USA) to each well. The wells were light-shielded and placed in an incubator for 2–3 h. We measured absorbance at 490 nm and 650 nm using a microplate reader ELx808 IU BioTek Instruments Inc. (Thermo Fisher Scientific, Walthum, MA, USA). Data were statistically analyzed using the Student’s *t*-test.

### 2.9. Scratch Assay

We used a scratch assay to evaluate cell migration ability. UE7T-9 cells were cultured at 3.0 × 10⁴ cells/70 µL using culture inserts (Ibidi, Gräfelfing, Germany). After 24 h, we removed the culture inserts, and HEK293T cells were cultured for 18 h. Formalin fixation and HE staining were then performed. The area between inserts after cell migration (cell-free area) was measured using Image J 1.53k (Wayne Rasband and contributors, National Institutes of Health, MD, USA). Data were statistically analyzed using the Student’s *t*-test.

### 2.10. Description of RNA Sequence Analysis

Total RNA was extracted from three samples, each of UE7T-9 SHC4 and UE7T-9 Cont, using a RNeasy Mini Kit (QIAGEN, Hilden, Germany). All library prep and whole genome sequencing were performed by Novogene Japan K.K. (Japan, https://jp.novogene.com/, accessed on 12 July 2022) using the NovaSeq 6000 Sequencing System (Illumina, CA, USA). Volcano plots were generated using Novosmart software (Novogene). Enrichment analysis was performed using clusterProfiler (Yu G, 2012) software based on Gene Ontology (GO), Disease Ontology (DO), Kyoto Encyclopedia of Genes and Genomes (KEGG), and Reactome data. For GSEA, normalized expression data were analyzed and visualized with GSEA software (version 4.2.3, https://www.gsea-msigdb.org/gsea/index.jsp, accessed on 27 May 2022). The normalized enrichment score (NES), nominal p-value (NOM-p), and false discovery rate q-value (FDR-q) were calculated for comparison, and the categories selected were universally upregulated in UE7T-9 cells. The relative enrichment of individual genes was assessed based on the rank metric score following the GSEA.

### 2.11. Statistical Analysis

The data were statistically analyzed using Graph Pad Prism Version 9.3.1 (350) software. The results were expressed as the mean ± standard deviation (SD). Statistical significance was determined by the Student’s *t*-test. A value of *p* < 0.05 was considered statistically significant for all analyses.

## 3. Results

### 3.1. Finding Candidate Factors for Intraosseous Progression Using CRISPR SAM Library

To identify candidate factors that could promote the infiltration of UE7T-9 cells into DCB, we introduced the CRISPR SAM activation library into UE7T-9 cells using a lentiviral vector and cultured them on DCB (Figure 1A). We also used HEK293T cells (human fetal kidney epithelial cells), which have a faster cell proliferation rate and easier gene transfer than UE7T-9 cells. After 3 weeks of culture, the UE7T-9 cells showed bone infiltration (Figure 1B), whereas the 293T cells showed no bone infiltration (Appendix A). The DNA sequence analysis of the infiltrated UE7T-9 cells identified five sequences. Among these, only one sequence matched the depositor data of the gRNA list of the CRISPR SAM library, which was a gRNA of *Src Homology 2 Domain Containing Family, Member 4 (**SHC4)*. The other four sequences did not match any gRNA in the library. CRISPR screening usually adjusts the MOI to a range of less than 1. However, in this study, we did not adjust the MOI because infiltration into the DCB was a priority; thus, we allowed multiple infections. Nevertheless, surprisingly, the only gRNA identified was *SHC4*. Therefore, we considered that *SHC4* overexpression is a defined factor that leads UE7T-9 cells into DCB.

### 3.2. Infiltration into DCB

Next, we created an overexpression vector for *SHC4* and *green fluorescent protein* (GFPGFP) as a control. By infecting UE7T-9 cells, we created *SHC4*-expressing cell lines (UE7T-9 SHC4) and GFPGFP-expressing cell lines (UE7T-9 Cont). Western blotting of these cell lines confirmed an *SHC4* overexpression in UE7T-9 SHC4 (Appendix A). To verify whether *SHC4* overexpression promotes intraosseous infiltration, we cultured UE7T-9 cells and UE7T-9 SHC4 on DCB for 8 weeks. A bone-infiltrating component was observed in UE7T-9 SHC4 (Figure 2A, Appendix A), whereas almost all cells were observed near the surface of the DCB in the case of UE7T-9 cells (Figure 2B, Appendix A). We measured the infiltrating distance from the DCB’s surface and observed a significant difference. (Figure 2C). This indicates that *SHC4* overexpression is an important factor in the infiltration of UE7T-9 cells into DCB.

### 3.3. Evaluation of Migration and Proliferation

#### 3.3.1. Scratch Assay Showing that *SHC4* Overexpression Enhanced the Migration Ability

We supposed that enhanced migration ability may be advantageous for infiltration into DCB. Thus, we used the scratch assay to assess whether *SHC4* overexpression is involved in migratory potential. After 18 h of incubation, active migration was observed, and the cell-free area showed a significant decrease in UE7T-9 SHC4 compared with the UE7T-9 control (*p* < 0.05) (Figure 3A–C). Thus, *SHC4* overexpression enhanced the migration of UE7T-9 cells.

#### 3.3.2. MTS Assay Showing There Are No Significant Difference in Proliferative Capacity

Since *SHC4* is a factor involved in cell proliferation signaling, such as *RAS* and *SRC,* it is important to evaluate cell proliferation. Furthermore, the possibility cannot be ruled out that the observed infiltration of the cells into the DCB may be due to an increase in proliferation. We used an MTS assay to evaluate whether cell proliferation had increased due to *SHC4* overexpression. A slight difference was observed in the proliferation of the UE7T-9 control compared to UE7T-9 SHC4 at 24 h of culture, but no significant difference was observed in the proliferation at 48 and 72 h of culture (Figure 3D). *SHC4* does not appear to be significantly involved in the proliferative capacity of UE7T-9 cells.

### 3.4. SHC4 Overexpression May Upregulate The Expression Of Genes Involved In Cell Migration

To evaluate gene expression changes along with *SHC4* overexpression, we performed an RNA sequence analysis. We extracted RNA from six samples, three each for UE7T-9 SHC4 and UE7T-9 Cont and performed an RNA sequence analysis. These data are accessible through the NCBI GEO database (http://www.ncbi.nlm.nih.gov/geo, accessed on 18 August 2022) with the accession number GSE209782. We examined the fold change in the expression of each gene in UE7T-9 SHC4 compared to UE7T-9 Cont and presented them as a volcano plot (Figure 4A). A total of 170 upregulated and 185 downregulated molecules were identified.

Further, GO enrichment analysis was performed to characterize the differentially expressed genes in the two groups. The analysis identified pathways related to hematopoietic cell migration, such as myeloid leukocyte migration, neutrophil migration, and granulocyte migration, as candidates (Figure 4B, Appendix A). Based on these results, we also performed a gene set enrichment analysis (GSEA) and found significant enrichment of related genes in UE7T-9 SHC4 compared to UE7T-9 Cont, particularly GOBP_Positive_Regulation_of_Blood_Vessel_Endothelial_Cell_Migration (Figure 4C). *STAT5A* was found in the gene sets (Appendix A). Several other gene sets were also identified (Appendix A). These results are consistent with the enhanced cell-migratory ability of UE7T-9 cells reinforced by *SHC4* overexpression.

## 4. Discussion

The introduction of hMSCs into DCB, a biomaterial, is very important for reconstructing the hybrid bone marrow niche. Despite various efforts in the past, the introduction of these cell lines has been difficult. Previous studies that have attempted to create models of the bone marrow niche have used primary hMSCs [12]. UE7T-9 cells are immortalized primary hMSCs introducing telomerase reverse transcriptase (hTERT) genes and human papillomavirus E7 genes. UE7T-9 cells are proliferative and can be easily transfected with lentiviral vectors. Further studies using other mesenchymal cell lines and primary hMSCs are necessary. In this study, we succeeded in introducing a human bone marrow mesenchymal cell line into DCB by using the CRISPR activation library. Notably, only *SHC4* was identified as a candidate gene for the infiltration of cells into DCB. Thus, we believe that CRISPR-activated libraries have high potential for application to the bone marrow niche, such as the introduction of cells into biomaterials, by supporting the easy and comprehensive upregulation of genes. Attempts to combine biomaterial with CRISPR for applications in disease modeling and regenerative medicine have begun to emerge in recent years [20,21], and this is the first study to apply the CRISPR library to dECM.

*SHC4* is a member of the *Src* gene family; *Src* family proteins function as adapter molecules for phosphotyrosine in a variety of receptor-mediated signaling pathways. The SHC family encodes at least six Shc-like proteins, characterized by an NH2-terminal phosphotyrosine-binding (PTB) domain, a central region rich in proline and glycine residues (CH1), and a COOH-terminal Src homologue 2 domain (SH2), in the presented order. *SHC4* is located on human chromosomes on 15q21.1–q21.2 and is specifically expressed during embryonic stem cell differentiation and early embryogenesis [22]. *SHC4* is expressed at high levels in adult brain tissue and at low levels in skeletal muscle (https://www.proteinatlas.org/ENSG00000185634-SHC4/tissue, accessed on 21 June 2022). In 2007, Ernesta Fagiani et al. identified *SHC4* (called RaLP at first) in vertically invading and metastasizing melanomas and showed that *SHC4* behaves as a substrate for IGF-1R or EGFR to promote melanoma cell migration [22]. Jones et al. showed that *SHC4* mediates signaling downstream of phosphorylated muscle-specific kinase (MuSK) receptor tyrosine kinase at the neuromuscular junction [23]. Melanie et al. found that *SHC4* promoted the autophosphorylation of EGFR in the absence of external stimuli. The authors used a quantitative PCR (qPCR) analysis of biopsy cores representing Grade I–IV astrocytomas to demonstrate the significantly higher expression of *SHC4* in tumor tissue compared to benign tissue [24,25]. Xin Zhang et al. showed that *SHC4* is expressed in hepatocellular carcinoma tissues and that *SHC4* promotes hepatocellular carcinoma progression by activating STAT3 signaling. They further noted that a high expression of *SHC4* is associated with clinicopathological features and poor prognoses in patients with hepatocellular carcinoma [24]. *SHC4* works as an oncogene and causes carcinoma cells to proliferate and become malignant. However, there have been few reports on *SHC4*, and its detailed function and how it works in normal cells, especially bone marrow mesenchymal cells, remain to be elucidated.

We evaluated the effect of *SHC4* overexpression on UE7T-9 cells. We found that *SHC4* overexpression did not promote cell proliferation but significantly enhanced cell migration in UE7T-9 cells. This was confirmed by the RNA sequence analysis, which showed an enrichment of genes involved in the pathway of cell migration in UE7T-9 SHC4 compared to the control, suggesting that *SHC4* overexpression enhanced the expression of genes that promote the migratory ability of UE7T-9 cells. We hypothesized that *SHC4* overexpression in UE7T-9 cells, inducing STAT5A upregulation, may lead to MSC migration [26].

*SHC4* upregulation in UE7T-9 cells has a minimal effect on cell morphology or proliferative potential. Given that *SHC4* does not involve major alterations in the properties of human bone marrow mesenchymal cells, it is a promising candidate for studying MDS caused by other genetic abnormalities, such as *DICER1,* in human mesenchymal cells. However, cell infiltration into DCB with *SHC4* overexpression was limited to DCB. In vivo transplantation experiments require a deeper penetration of cells into the bone marrow matrix. In the future, we will consider using a third-generation CRISPR activation library, such as the Calabrese library, or using the CRISPR knockout library to search for other infiltrating factors. We believe that the successful infiltration and engraftment of hMSCs into DCB will provide a milestone for recapitulating the bone marrow niche. If the human–mouse hybrid bone marrow niche can be reproduced in mice, it will be easier to create models of MDS by inducing specific genetic abnormalities, such as *DICER1,* in hMSCs. Transplanted DCBs are easy to recollect and analyze the tumorigenesis of MDS for each time course.

## Figures and Tables

**Figure 1 bioengineering-09-00490-f001:**
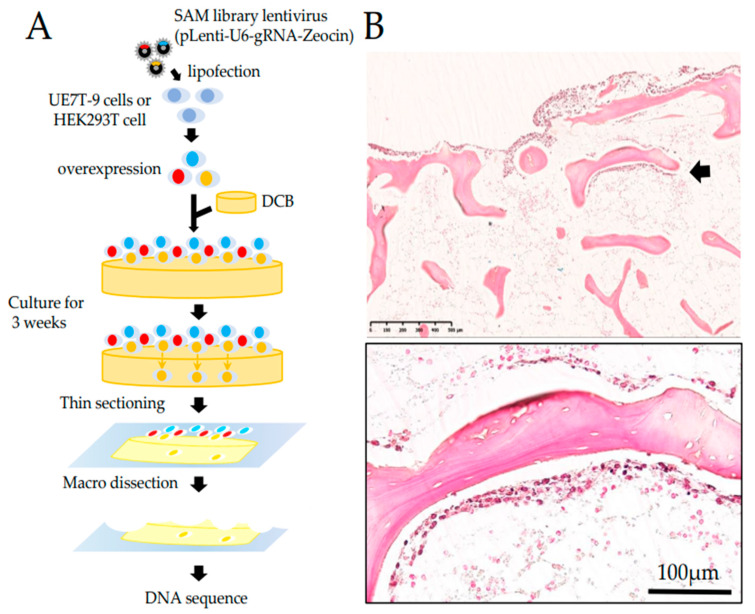
Finding candidate factors of intraosseous progression using CRISPR SAM library. (**A**) Schematic strategy for finding candidate factors. We introduced the CRISPR SAM library into the cell lines to randomly induce gene expression. We cultured these cells on DCB for 3 weeks and performed DNA sequencing analysis on the cellular components that infiltrated into DCB. (**B**) HE-stained specimen prepared after 3 weeks of culture. The UE7T-9 cells, expressing the CRISPR-SAM library, infiltrated along the bone trabeculae (black arrow). The bottom figure is a high-power field of the above.

**Figure 2 bioengineering-09-00490-f002:**
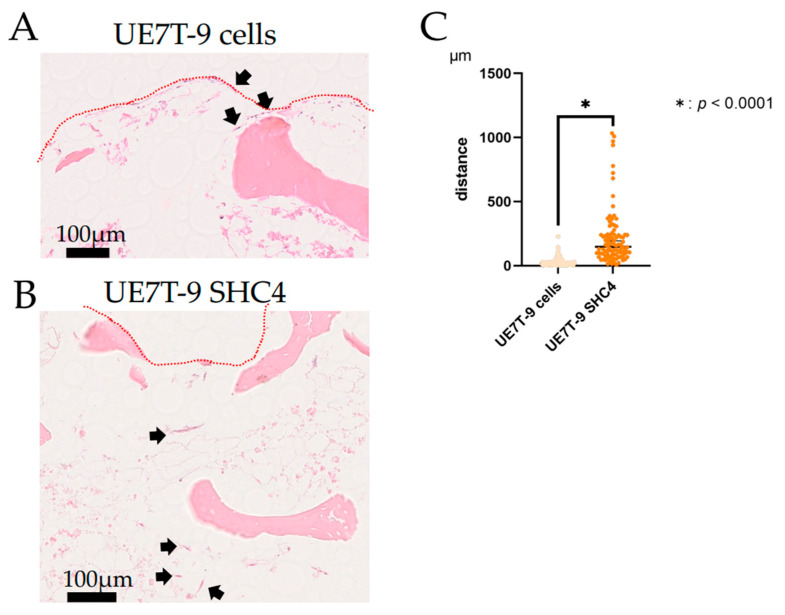
Infiltration into DCB. (**A**) HE-stained specimen of DCB, after 8-weeks of culture over UE7T-9 cells. UE7T-9 cells remained on the surface of the DCB. (**B**) HE-stained specimen of DCB after 8-weeks of culture of UE7T-9 SHC4. There were infiltrating cells around the bone trabecular (black arrow). (**C**) The infiltrating distance measured in UE7T-9 cells (*n* = 117) and UE7T-9 SHC4 cells (*n* = 96). The cells were randomly selected for distance measurement. Graph showing the distance that UE7T-9 cells infiltrated from the surface of DCB (red dot line). * indicates a significant difference of *p* < 0.0001 between the results of the two groups.

**Figure 3 bioengineering-09-00490-f003:**
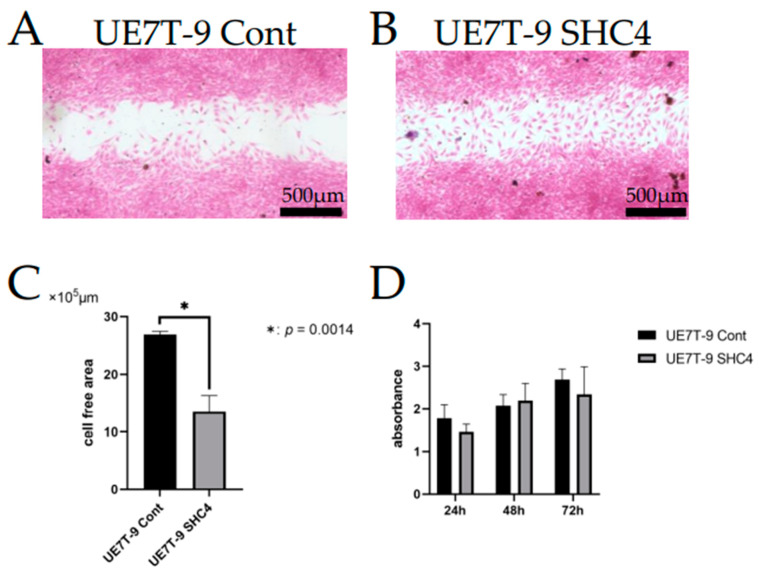
Evaluation of migration and proliferation. (**A**,**B**) Photographs of HE staining of UE7T-9 cells after 18 h of culture in scratch assay (A: UE7T-9 Cont B: UE7T-9 SHC4). (**C**) The graph shows the cell-free area of the scratched UE7T-9 cell culture after 18 h of culture, as measured by Image J. The cell-free area of UE7T-9 SHC4 was significantly decreased compared to the control (*p* < 0.05), indicating enhanced migration due to *SHC4* overexpression. (**D**) Evaluation of proliferative potential by MTS assay. No significant difference in proliferation was observed between the UE7T-9 control and the UE7T-9 SHC4.

**Figure 4 bioengineering-09-00490-f004:**
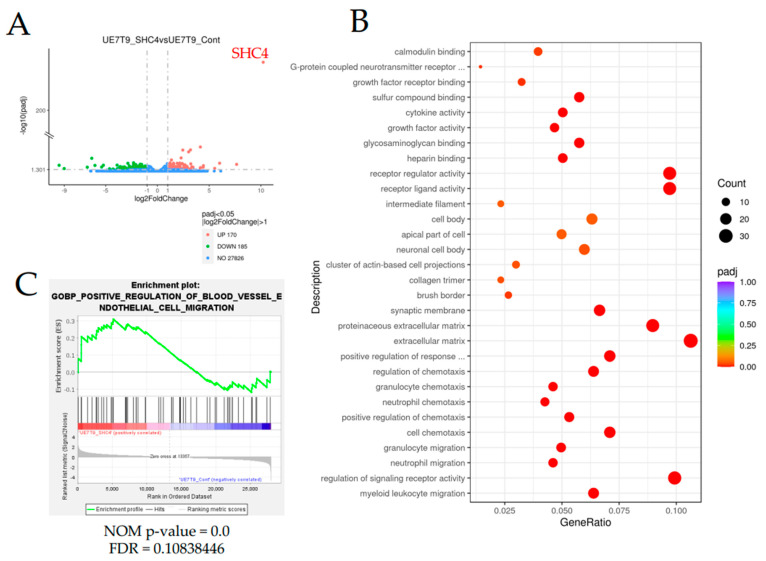
Results of RNA sequence analysis of UE7T-9 cell lines with increased *SHC4* gene expression. (**A**) Volcano plot showing the distribution of genes expressed in UE7T-9 SHC4 according to their expression levels when compared to UE7T-9 Cont. The x-axis (log2FoldChange) shows the fold change in gene expression between different samples, and the y-axis (-log10(padj)) shows the statistical significance of the difference. Red dots represent upregulated genes, and green dots represent downregulated genes. (**B**) Pathway enrichment analysis shows the pathways to which the differential genes are most related, with reference to the Gene Ontology database. (**C**) Graphical views of the enrichment plot of the GSEA results for GOBP_Positive_Regulation_of_Blood_Vessel_Endothelial_Cell_Migration.

## Data Availability

Not applicable.

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
