# Peer review of "Identification of the Factor That Leads Human Mesenchymal Stem Cell Lines into Decellularized Bone"

_bioengineering, 2022, doi:10.3390/bioengineering9100490_

Round 1
Reviewer 1 Report
This manuscript concludes that SHC4 is a key gene for engrafting lined MSC UE7T-9 into DCB.
I think this manuscript is well organized and I can understand the key factor for leading MSC into DCB.
However, I have some questions and requests as follows:
1. The authors have focused on SHC4, but I would like you to list the other candidates and state detailly the reason.
2. The authors have mentioned the difficulty of primary hMSC induction into DCB. I wonder why primary hMSC cannot be induced but UE7T-9 can be induced. Please state the reason with a little detailed mechanism or regulatory genes in the Discussion section.
3. For Fig4, please list up most important gene you think of and state the reason or hypothesis in the DISCUSSION section.
4. All histological photos and Fig4 data are a little unclear and unfocused. Please change them to more fine ones.
Author Response
Authors want to thank the reviewers for their thoughtful and thorough analysis of our manuscript. We have endeavored to address all of the issues they presented. The reviewers’ comments are in bold, and the author’s response is indicated in blue.
This manuscript concludes that SHC4 is a key gene for engrafting lined MSC UE7T-9 into DCB.
I think this manuscript is well organized and I can understand the key factor for leading MSC into DCB. However, I have some questions and requests as follows:
- The authors have focused on SHC4, but I would like you to list the other candidates and state detailly the reason.
Response: Thank you for your pointing this out. The DNA sequence analysis of the infiltrated UE7T-9 cells identified five sequences. Among these, only 1 sequence matched the gRNA list of the CRISPR SAM library, which was a gRNA of SHC4. That was the reason we focused on SHC4. We have added the description.
- The authors have mentioned the difficulty of primary hMSC induction into DCB. I wonder why primary hMSC cannot be induced but UE7T-9 can be induced. Please state the reason with a little detailed mechanism or regulatory genes in the Discussion section.
Response: Thank you for your suggestion. UE7T-9 cells are immortalized primary hMSCs introducing human telomerase reverse transcriptase (hTERT) genes and human papillomavirus E7 genes. UE7T-9 cells proliferate and are easy to introduce into lentiviral vector, so that is why we used UE7T-9 cells. We think SHC4-overexpression in UE7T-9 cells is a key factor for induction into DCB because SHC4-overexpression leads cell motility and upregulates hematopoietic cell migration factors, based on our data. We have not tried experiments with primary hMSCs, but it is important to perform experiments using primary cells. We have added the description to the Discussion section.
- For Fig4, please list up most important gene you think of and state the reason or hypothesis in the DISCUSSION section.
Response: Thank you for bringing up the issue. In GSEA, signal transducer and activator of transcription 5A (STAT5A) was one of the top genes involved in the focused pathway. We hypothesized that SHC4 overexpression in MSCs, inducing STAT5A upregulation, may lead to MSC migration. In the GO pathway enrichment analysis, migration pathways were on the top, and Lipopolysaccharide binding proteins (LBP) were commonly included among the genes involved in the migration pathway. We considered these two genes to be more important among the genes evaluated in this RNA sequence analysis. We also evaluated UE7T-9 SHC4 and UE7T-9 Cont by qRT-PCR, which supported that both STAT5A and LBP are upregulated. qRT-PCR results are attached below. We changed the DISCUSSION section as follows.
--Changed script--
We hypothesized that SHC4 overexpression in UE7T-9 cells, inducing STAT5A upregulation, may lead to MSC migration
- All histological photos and Fig4 data are a little unclear and unfocused. Please change them to more fine ones.
Response: We replaced the image with a clearer one and adjusted the size.

Reviewer 2 Report
In this paper, Koyanagi and colleagues identified, by the use of a CRISPR activation library, SHC4 as a gene related to the capacity of the stromal cell line UE7T-9 to migrate into decellularized bone (DCB) and to engraft the scaffold. The aim of the work is to generate a system using DCB to recreate a humanized bone marrow niche in vivo. The work is potentially interesting, also if the results seem to be very partial, but the text needs strong editing. For example, a large part of the introduction was dedicated to the role of mutations of MSC in MDS predisposition, but this is not the specific focus of the work. This becomes confusing in my opinion.
In general, I suggest extensive editing of English language and style.
- Could you provide an image of "empty DBC" to be sure that infiltrating cells are UE7T-9 SHC4 and not pre-existent cells resisting hydrostatic pressurization?
- Also in the case of UE7T-9 SHC4, you found very few cells within DBC. Do you think that they can be enough for your final goal (the creation of a humanized niche)?
- What do you mean by "from each of the three samples of UE7T-9" (page 8, line 299)? Three independent cell lines?
- Figure 3D: you mentioned in the text (page 7, line 286) that a slight difference was observed in the proliferation between SHC4 and CTR at 24h but no asterisks representing significance were reported in the graph. Be careful, you wrongly referred to Fig 3C and not to 3D. Please clarify.
- You found by RNAseq analysis some pathways upregulated in UE7T-9 SHC4 specifically involved in migration. Have you validated some of these results?
- Why do you think that the use of DBC can be superior to the already existing humanized BM niche models? Please clarify.
Author Response
Authors want to thank the reviewers for their thoughtful and thorough analysis of our manuscript. We have endeavored to address all of the issues they presented. The reviewers’ comments are in bold, and the author’s response is indicated in blue.
In this paper, Koyanagi and colleagues identified, by the use of a CRISPR activation library, SHC4 as a gene related to the capacity of the stromal cell line UE7T-9 to migrate into decellularized bone (DCB) and to engraft the scaffold. The aim of the work is to generate a system using DCB to recreate a humanized bone marrow niche in vivo. The work is potentially interesting, also if the results seem to be very partial, but the text needs strong editing. For example, a large part of the introduction was dedicated to the role of mutations of MSC in MDS predisposition, but this is not the specific focus of the work. This becomes confusing in my opinion.
Response: Thank you for pointing this out. As you mentioned, the MDS description took up a large area of the introduction, so we made this section simpler. We are concerned with MSCs because they are important components of the niche, and genetic mutations in MSCs can be induced to create specific disease models. We also wanted to put the disease model in a form that could be used in vivo in animal experiments. Our ultimate goal at present is to create a disease model of MDS.
In general, I suggest extensive editing of English language and style.
- Could you provide an image of "empty DBC" to be sure that infiltrating cells are UE7T-9 SHC4 and not pre-existent cells resisting hydrostatic pressurization?
Response: As you mentioned, showing an empty DCB histology is very important. We added that image and the description in Supplementary Fig. 1. Pre-existent cells died and degenerated under hydrostatic pressurization.
- Also in the case of UE7T-9 SHC4, you found very few cells within DBC. Do you think that they can be enough for your final goal (the creation of a humanized niche)?
Response: You are correct that the amount of UE7T-9 cells infiltrating the DCB is insufficient to form a bone marrow niche. As noted in the discussion, we would like to use other tools, such as other CRISPR knock-out libraries, to look for more factors that lead hMSCs into DCB.
- What do you mean by "from each of the three samples of UE7T-9" (page 8, line 299)? Three independent cell lines?
Response: That statement intended to say three samples each for UE7T-9, SHC4, and UE7T-9 Cont. The words were misleading, so we have corrected them.
- Figure 3D: you mentioned in the text (page 7, line 286) that a slight difference was observed in the proliferation between SHC4 and CTR at 24h but no asterisks representing significance were reported in the graph. Be careful, you wrongly referred to Fig 3C and not to 3D. Please clarify.
Response: Thank you for pointing out this error. We have corrected it.
- You found by RNAseq analysis some pathways upregulated in UE7T-9 SHC4 specifically involved in migration. Have you validated some of these results?
Response: Thank you for bringing up the issue. In GSEA, signal transducer and activator of transcription 5A (STAT5A) was one of the top genes involved in the focused pathway. We hypothesized that SHC4 overexpression in MSCs, inducing STAT5A upregulation, may lead to MSC migration. In the GO pathway enrichment analysis, migration pathways were on the top, and Lipopolysaccharide binding proteins (LBP) were commonly included among the genes involved in the migration pathway. We considered these two genes to be more important among the genes evaluated in this RNA sequence analysis. We also evaluated UE7T-9 SHC4 and UE7T-9 Cont by qRT-PCR, which supported that both STAT5A and LBP are upregulated. qRT-PCR results are attached below. We changed the DISCUSSION section as follows.
--Changed script--
We hypothesized that SHC4 overexpression in UE7T-9 cells, inducing STAT5A upregulation, may lead to MSC migration
- Why do you think that the use of DBC can be superior to the already existing humanized BM niche models? Please clarify.
Response: As noted in the introduction, Hashimoto et al. reported that subcutaneous transplantation of DCBs into rats established hematopoietic tissue autonomously in DCBs without provoking an immune response [7]. There were CD34-positive hematopoietic stem cells. Most other scaffolds have utilized a combination of collagen, poly-hydroxyapatite (HA), and beta-tricalcium phosphate (b-TCP) for cell trafficking, but DCB may have signal molecules remaining on the decellularized, extracellular matrix that induced hematopoiesis. Based on these results, we selected DCBs as a scaffold.

Reviewer 3 Report
The current manuscript describes a very interesting study that aimed at finding a way to populate decellularized bone with hMSCs.
Introduction is well structured and comprehensive, and familiarizes the reader with the background of the study, while also providing a strong motivation for the need of this research and also pointing out the novelty and added value that it brings to current knowledge.
The study design is complex and appropriate, able to demonstrate the working hypothesis. Materials and methods are well described and executed, presented in enough detail to make them reproducible.
Results are very clear, the figures are of good quality and are representative, while the supplementary material is also useful and very welcome. Statistical analysis is adequate and sufficient.
Discussions are very well conducted and emphasize again on the originality and added value of the study. The literature cited is relevant and well chosen.
English language is adequate, the information is very well organized, in a logical manner and easy to follow.
The method described here is original and innovative, and brings an important step forward towards the use of bone marrow niche in future research, related to MDS as well as other diseases. This manuscript is very valuable and I have no objections towards its publication as it is.
Author Response
Authors want to thank the reviewers for their thoughtful and thorough analysis of our manuscript. We have endeavored to address all of the issues they presented. The reviewers’ comments are in bold, and the author’s response is indicated in blue.
The current manuscript describes a very interesting study that aimed at finding a way to populate decellularized bone with hMSCs.
Introduction is well structured and comprehensive, and familiarizes the reader with the background of the study, while also providing a strong motivation for the need of this research and also pointing out the novelty and added value that it brings to current knowledge.
The study design is complex and appropriate, able to demonstrate the working hypothesis. Materials and methods are well described and executed, presented in enough detail to make them reproducible.
Results are very clear, the figures are of good quality and are representative, while the supplementary material is also useful and very welcome. Statistical analysis is adequate and sufficient.
Discussions are very well conducted and emphasize again on the originality and added value of the study. The literature cited is relevant and well chosen.
English language is adequate, the information is very well organized, in a logical manner and easy to follow.
The method described here is original and innovative, and brings an important step forward towards the use of bone marrow niche in future research, related to MDS as well as other diseases. This manuscript is very valuable and I have no objections towards its publication as it is.
Thank you for the review and kindful message.
We are very impressed with the high regard you have given to our paper.

Round 2
Reviewer 2 Report
The authors replied to all the requests. However, the references to existent models of humanized bone marrow niches using human MSCs are very partial (ref 12). Please add at least some review (e.g. Abarrategi et al JEM 2018; Dupard SJ et al Trends in Molecular Medicine 2020;...). Please discuss also in the text the reason for the potential superiority of DCB over the classical scaffolds.
Author Response
Reviewer’s comment
The authors replied to all the requests.
However, the references to existent models of humanized bone marrow niches using human MSCs are very partial (ref 12). Please add at least some review (e.g. Abarrategi et al JEM 2018; Dupard SJ et al Trends in Molecular Medicine 2020;...). Please discuss also in the text the reason for the potential superiority of DCB over the classical scaffolds.
Response
Thank you very much for sharing these informative studies with us. We have gained insight into the humanized bone marrow model in mice and humanized ossicles. We should have mentioned the reason for the superiority of DCB over other scaffolds. We have added references and revised the INTRODUCTION section as follows.
We would like to use other humanized bone marrow niche models in the next step.
